# Safety and Effectiveness of Abatacept in a Prospective Cohort of Patients with Rheumatoid Arthritis–Associated Interstitial Lung Disease

**DOI:** 10.3390/biomedicines10071480

**Published:** 2022-06-22

**Authors:** Natalia Mena-Vázquez, Marta Rojas-Gimenez, Clara Fuego-Varela, Aimara García-Studer, Nair Perez-Gómez, Carmen María Romero-Barco, Francisco Javier Godoy-Navarrete, Sara Manrique-Arija, Myriam Gandía-Martínez, Jerusalem Calvo-Gutiérrez, Pilar Morales-Garrido, Coral Mouriño-Rodriguez, Patricia Castro-Pérez, Isabel Añón-Oñate, Francisco Espildora, María Carmen Aguilar-Hurtado, Ana Hidalgo Conde, Rocío Arnedo Díez de los Ríos, Eva Cabrera César, Rocío Redondo-Rodriguez, María Luisa Velloso-Feijoo, Antonio Fernández-Nebro

**Affiliations:** 1Instituto de Investigación Biomédica de Málaga (IBIMA), 29010 Malaga, Spain; aimara.garcia.sspa@juntadeandalucia.es (A.G.-S.); carmenm.romero.barco.sspa@juntadeandalucia.es (C.M.R.-B.); sara.manrique@uma.es (S.M.-A.); rocio.redondo.sspa@juntadeandalucia.es (R.R.-R.); afernandezn@uma.es (A.F.-N.); 2UGC de Reumatología, Hospital Regional Universitario de Málaga, 29009 Malaga, Spain; 3Instituto Maimónides de Investigación Biomédica de Córdoba (IMIBIC), 14004 Cordova, Spain; marta.rojas.gimenez.sspa@juntadeandalucia.es (M.R.-G.); yeru83@hotmail.com (J.C.-G.); 4UGC de Reumatología, Hospital Universitario Reina Sofía de Córdoba, 14004 Cordova, Spain; 5UGC de Reumatología, Hospital Universitario de Jerez, 11407 Cadiz, Spain; clara.fuego.sspa@juntadeandalucia.es (C.F.-V.); myriamgandia@hotmail.com (M.G.-M.); 6UGC de Reumatología, Complejo Hospitalario Universitario de Vigo, 36214 Vigo, Spain; nair.perez.gomez@sergas.es (N.P.-G.); coral.mourino@iisgaliciasur.es (C.M.-R.); 7UGC de Reumatología, Hospital Clínico Universitario Virgen de la Victoria, 29010 Malaga, Spain; 8UGC de Reumatología, Hospital Universitario de Jaén, 23007 Jaen, Spain; francisco.godoy.navarrete.sspa@juntadeandalucia.es (F.J.G.-N.); isaanononate@gmail.com (I.A.-O.); 9Departamento de Medicina, Universidad de Málaga, 29010 Malaga, Spain; 10UGC de Reumatología, Hospital Universitario Clínico San Cecilio, 18016 Granada, Spain; pilar.morales.sspa@juntadeandalucia.es; 11UGC de Reumatología, Hospital Universitario de Getafe, 28901 Madrid, Spain; patricastroperez@gmail.com; 12UGC de Neumología, Hospital Regional Universitario de Málaga, 29009 Malaga, Spain; fespildorahernandez@gmail.com; 13UGC de Radiodiagnóstico, Hospital Regional Universitario de Málaga, 29009 Malaga, Spain; maguh007@gmail.com; 14Servicio de Medicina Interna, Hospital Universitario Virgen de la Victoria, 29010 Malaga, Spain; ana.hidalgo.conde.sspa@juntadeandalucia.es (A.H.C.); rocioardiez@gmail.com (R.A.D.d.l.R.); 15UGC Neumología, Hospital Universitario Virgen de la Victoria, 29010 Malaga, Spain; evacabreracesar@gmail.com; 16UGC de Reumatología, Hospital Universitario Virgen de Valme, 41014 Sevilla, Spain; mlvelloso@hotmail.com

**Keywords:** rheumatoid arthritis, interstitial lung disease, biologics, abatacept

## Abstract

Objective: To prospectively evaluate the safety and efficacy profile of abatacept in patients with rheumatoid arthritis–associated interstitial lung disease (RA-ILD). Methods: We performed a prospective observational multicenter study of a cohort of patients with RA-ILD treated with abatacept between 2015 and 2021. Patients were evaluated using high-resolution computed tomography and pulmonary function tests at initiation, 12 months, and the end of follow-up. The effectiveness of abatacept was evaluated based on whether ILD improved, stabilized, progressed, or was fatal. We also evaluated factors such as infection, hospitalization, and inflammatory activity using the 28-joint Disease Activity Score with the erythrocyte sedimentation rate (DAS28-ESR). Cox regression analysis was performed to identify factors associated with progression of lung disease. Results: The study population comprised 57 patients with RA-ILD treated with abatacept for a median (IQR) of 27.3 (12.2–42.8) months. Lung disease had progressed before starting abatacept in 45.6% of patients. At the end of follow-up, lung disease had improved or stabilized in 41 patients (71.9%) and worsened in 13 (22.8%); 3 patients (5.3%) died. No significant decreases were observed in forced vital capacity (FVC) or in the diffusing capacity of the lung for carbon monoxide (DLCO).The factors associated with progression of RA-ILD were baseline DAS28-ESR (OR [95% CI], 2.52 [1.03–3.12]; *p* = 0.041), FVC (OR [95% CI], 0.82 [0.70–0.96]; *p* = 0.019), and DLCO (OR [95% CI], 0.83 [0.72–0.96]; *p* = 0.018). Only 10.5% of patients experienced severe adverse effects. Conclusion: Pulmonary function and joint inflammation stabilized in 71% of patients with RA-ILD treated with abatacept. Abatacept had a favorable safety profile.

## 1. Introduction

Rheumatoid arthritis (RA) is a chronic disease of unknown cause that mainly affects the joints, although extra-articular clinical manifestations are frequent [1,2]. The lung is one of the most affected organs, leading to significant morbidity and mortality [3,4]. While around 20–30% of patients develop clinically significant rheumatoid arthritis–associated interstitial lung disease (RA-ILD), systematic screening has shown that up to 35–50% of patients with established RA develop the disease [5]. The factors associated with a poorer prognosis in patients with RA-ILD include advanced age, male sex, antibody levels, poor control of previous joint inflammation, impaired lung function at diagnosis, and a usual interstitial pneumonia (UIP) pattern on high-resolution computed tomography (HRCT) [6,7,8,9,10,11,12,13,14,15].

Immunosuppressants such as mycophenolate mofetil, azathioprine, and cyclophosphamide are useful for the treatment of RA-ILD, although they have little effect on joint involvement [16,17,18]. The same is true of antifibrotic agents, as shown in the INBUILD study [19]. In contrast, the conventional synthetic disease-modifying antirheumatic drugs (csDMARDs) and biologic disease-modifying antirheumatic drugs (bDMARDs) administered for control of RA have proven highly useful for joint symptoms and, according to recent studies, can also help to stabilize or delay progression of RA-ILD [20,21,22]. While some studies report an association between methotrexate and ILD, other, more recent, and higher-quality studies do not confirm this association [23,24]. Evidence is scarcer for the remaining csDMARDs, although one meta-analysis revealed more frequent respiratory adverse effects [22]. As for bDMARDs, available evidence, which is based mainly on cross-sectional and retrospective studies, suggests that rituximab and abatacept could be the safest drugs for the treatment of RA-ILD [20,21,25,26,27,28], whereas anti-tumor necrosis factor blockers (anti-TNF) may be associated with a risk of worsening of pre-existing RA-ILD [22].

Abatacept is a fusion protein formed by the extracellular portion of the cytotoxic T lymphocyte-associated antigen 4 (CTLA-4) and the Fc fragment of human IgG1. It has been approved for treatment of RA, the polyarticular form of juvenile idiopathic arthritis, and psoriatic arthritis [29]. Recent data suggest that it can stabilize and, in some cases, improve lung function in patients with RA-ILD, although published studies on this subject are mainly retrospective and few in number. Therefore, based on a prospective multicenter RA-ILD registry [21,28], we evaluated patients who initiated abatacept with the following objectives: (1) to prospectively evaluate the effectiveness and safety of abatacept for treatment of RA-ILD in clinical practice; and (2) to identify risk factors that will help us to predict disease progression in patients treated with abatacept.

## 2. Materials and Methods

### 2.1. Design

We performed a multicenter, prospective, observational study of a series of patients with RA-ILD receiving abatacept nested in a prospective cohort of 11 teaching hospitals in Spain. The study was approved by the Ethics Committee of Hospital Regional Universitario de Málaga (HRUM) (Code 1719-N-15). All the patients participated voluntarily and provided their written informed consent before entering the study.

### 2.2. Study Population

Patients with RA-ILD who were candidates for abatacept were recruited at the participating sites between March 2015 and December 2021. ILD was confirmed by pulmonary function testing (PFT) and HRCT or lung biopsy. The eligibility criteria were age ≥ 18 years, RA classified according to the 2010 criteria of the ACR/EULAR [30], and ≥6 months’ treatment with abatacept. We excluded patients with inflammatory or rheumatic diseases other than RA (except secondary Sjögren syndrome), lung involvement other than ILD, active infection, pulmonary hypertension, and heart disease.

### 2.3. Protocol

We prospectively followed a cohort of patients with RA-ILD recruited consecutively between 2015 and 2021 from the rheumatology clinic at various centers throughout Spain. The population of the present study comprised patients who initiated treatment with abatacept during the prospective follow-up period. Patients were evaluated using HRCT and PFT at the initiation of treatment (V0), at 12 months (V12), and at the end of follow-up in 2021 (final visit (FV)). All HRCT scans were based on an axial slice measuring 1.5 or 2 mm in thickness taken at intervals of 1 cm along the thorax. Images were reconstructed using a high-spatial-frequency-algorithm, with 20 to 25 slices acquired per patient per minute. we have performed a visual method to evaluate the extension of ILD and the parenchymal pattern score was calculated based on the percentage ratio of the area of each parenchymal pattern to the total lung parenchyma [31,32,33]. In order to homogenize the interpretation of findings, the radiological evaluation was centralized at HRUM and performed blind and independently by two experts in pulmonary radiology.

All patients started treatment with subcutaneous abatacept at 125 mg/week (V0). Patients were then evaluated every 3–6 months by a rheumatologist and every 6–12 months by a pulmonologist. All data were collected according to a pre-established protocol.

### 2.4. Working Definitions and Variables

The main variable was the effectiveness of abatacept according to the outcome of ILD at the end of follow-up (FV) with respect to the following: (1) improvement (i.e., improvement in forced vital capacity (FVC) ≥ 10% or in the diffusing capacity of the lung for carbon monoxide (DLCO) ≥ 15% and no radiological progression); (2) non progression (stabilization or improvement in FVC < 10% or in DLCO < 15% and no radiological progression); (3) progression (worsening of FVC > 10% or DLCO > 15% or radiological progression); and (4) death.

Similarly, as secondary variables, we evaluated the change in FVC and DLCO, and radiologic progression at 12 months of treatment with abatacept (V12) and at the end of follow-up (FV). The lung function was considered progression (decrease in forced vital capacity (FVC) >10% or in the diffusing capacity of the lung for carbon monoxide (DLCO) > 15), stabilization (no progression or increase in FVC < 10% or in DLCO < 15%) and improvement (FVC ≥ 10% or DLCO ≥ 15%) [34,35]. Radiologic evaluation was considered progression (≥20% increase in the presence and extension of ground-glass opacities, reticulation, honeycombing, diminished attenuation, centrilobular nodules, other nodules, emphysema, and consolidation), stabilization (extension < 20%), and improvement (decrease in extension) [31,32,33,36].

ILD was defined based on lung biopsy or HRCT findings according to the standard criteria of the American Thoracic Society/European Respiratory Society International Multidisciplinary Consensus Classification of the Idiopathic Interstitial Pneumonias [37] and classified into three patterns: nonspecific interstitial pneumonia (NSIP), UIP, and other (bronchiolitis obliterans, organizing pneumonia, lymphocytic interstitial pneumonitis, and mixed patterns). PFT included full spirometry, where the results were expressed as percent predicted adjusted for age, sex, and height. Abnormal FVC was defined as <80% predicted. DLCO was evaluated using the single-breath method (DLCO-SB), with a value of <80% considered abnormal [38].

Other variables included duration of joint symptoms, diagnostic delay, smoking history (current or previous), and body mass index (weight/height^2^). The joint evaluation at initiation of abatacept and at the end of follow-up included inflammatory activity measured using the 28-joint Disease Activity Score with erythrocyte sedimentation rate (DAS28-ESR) (range, 0–9.4) [39] and physical function based on the Health Assessment Questionnaire [40]. Other inflammation-related variables included blood C-reactive protein (mg/dL) and ESR (mm/h). At V0, we also collected variables associated with severity: rheumatoid factor (reference value, 20 U/mL; high titer, >60 U/mL); anticitrullinated peptide antibody (ACPA) (reference value, 10 U/mL; high value, >340 U/mL); and presence of at least 1 radiologic erosion. We recorded concomitant treatment with csDMARDs (methotrexate, leflunomide, sulfasalazine, and hydroxychloroquine), immunosuppressants (mycophenolate mofetil, azathioprine), antifibrotic agents (both previous and during follow-up), and corticosteroids. We reported discontinuation of abatacept and reasons for discontinuation. As for adverse events, we reported infections, hospitalization as a clinical event, and reason for hospitalization [41].

### 2.5. Statistical Analysis

We performed a descriptive analysis of the clinical, laboratory, and therapy-related characteristics of patients with RA-ILD receiving abatacept. Qualitative variables were expressed as a whole number and percentage, and quantitative variables were expressed as mean and standard deviation (SD) or as median and interquartile range (IQR) depending on the normality of their distribution according to the Kolmogorov–Smirnov test. The bivariate analysis was performed using the paired *t* or Wilcoxon test, as applicable, between V0 and V12 and between V0 and FV. Kaplan–Meier curves were used to estimate the survival of patients with RA-ILD receiving abatacept. Survival time was measured from V0 until FV or death. Cox regression analysis was applied to identify prognostic factors for time to progression or death using a univariate model and a multivariate model (forward stepwise). All variables that reached a *p* value of <0.10 were included in the Cox multivariate model. The analysis was carried out using R Commander.

## 3. Results

### 3.1. Baseline Clinical Characteristics

Between March 2015 and September 2021, a total of 59 patients with RA-ILD in prospective follow-up initiated abatacept, although 57 patients with a follow-up of ≥6 months were eventually included (Figure 1). The median (IQR) time with abatacept was 27.3 (12.2–42.8) months. The main baseline characteristics (V0) are shown in Table 1. Patients were aged around 67 years and were equally distributed by sex. One third of the patients had been smokers or were smokers at inclusion, and almost all of them had longstanding seropositive disease. As can be seen in Table 1, the mean DAS28-ESR at initiation of abatacept indicated moderate inflammatory activity, and ILD had been progressing for a median of 4 years.

At V0, 47/57 patients (82.5%) were receiving abatacept combined with csDMARDs, 5/57 (8.8%) were receiving abatacept combined with an immunosuppressant, and 5/57 (8.8%) were receiving abatacept in monotherapy. Only 1 patient was receiving abatacept combined with mycophenolate mofetil and nintedanib. The DMARDs prescribed at V0 are shown in Table 1. More than half of the patients in the study were taking corticosteroids at a mean (IQR) dose of 5.0 mg/d (2.0–5.0). Forty-three patients (75.4%) had received at least 1 csDMARD before V0, 21 (36.8%) had previously received a bDMARD for a median of 3.2 months (16.0–54.7), and 5 (6.8%) had received an immunosuppressant (Appendix A). Abatacept was the first biologic prescribed in 36/57 patients (63.1%).

The most common radiologic pattern was UIP (34/57 patients (59.6%)), followed by NSIP (18/57 (31.6%)) and fibrotic NSIP (5/57 (8.8%)). Three patients had histology-confirmed UIP-type ILD.

### 3.2. Progression of Lung Disease

At the end of follow-up, almost three-quarters of the patients had improved or stabilized (72%), and 28% had worsened or died (Table 2). The median (95% CI) survival until progression or death was 72.0 months (34.0–110.0) (Figure 2).

At initiation of abatacept, mean PFT values had decreased significantly compared with the date of diagnosis of ILD, both for FVC (mean (SD), 76.5 (18.2) vs. 80.4 (13.6) mg/L; *p* = 0.042) and for DLCO-SB (mean (SD), 62.5 (16.5) vs. 68.4 (16.2) mg/L; *p* = 0.032) (Appendix A). In addition, as can be seen in Table 2, the progression of lung disease prior to starting abatacept was more marked than during the period of treatment with abatacept (45.6% vs. 22.8%; *p* = 0.019). The mean PFT values did not decrease significantly during the first 12 months of abatacept compared with the baseline values or at the end of follow-up (Figure 3). Appendix A shows the PFT values, progression on HRCT, and overall progression of lung disease from baseline, as well as progression at 12 months and at the end of follow-up. At the end of follow-up, radiologic progression on HRCT was observed in 13/57 patients (22.8%), function pulmonary progression for Pulmonary function tests in 13/57 patients (22.8%), and 3/57 patients (5.3%) had died.

Among the 26 (46%) patients who had previously progressed prior to abatacept, 13/26 (50%) had progression at the end of follow-up and another 13/26 (50%) had no progression (*p* = 1.000), whereas among patients whose disease had been stable before abatacept, 3/31 (9.7%) progressed and N28/31 (90.3%) did not progress (*p* = 0.001).

Compared with initiation of abatacept, joint inflammation improved significantly at the end of follow-up in terms of DAS28-ESR (mean (DS), 3.1 (1.2) vs. 3.8 (1.5) mg/L; *p* = 0.024), CRP (median (IQR), 3.3 (2.6–7.9) vs. 7.0 (5.0–17.7) mg/L; *p* = 0.018), and ESR (median (IQR), 20.0 (8.0–29.0) vs. 33.0 (15.0–47.0) mg/L; *p* = 0.046). In addition, fewer patients were taking corticosteroids at the end of follow-up compared with at the initiation of abatacept (48.4% vs. 68.4%; *p* = 0.040).

### 3.3. Adverse Events

The main adverse events are shown Table 3. A total of 25/57 patients (43.9%) developed an infection during the follow-up period, and 6/57 patients (10.5%) were admitted to hospital at least once. The most frequent infections were respiratory infection (35.0%), followed by urinary infection (7.0%). Three patients were admitted to hospital and died owing to progression of ILD.

Eight of 57 patients (14.0%) discontinued treatment during follow-up: 3 (5.2%) did so owing to progression of lung disease, 4 (7.0%) owing to joint failure, and 1 (1.8%) owing to recurrent urinary infection and joint failure. Appendix A shows follow-up times, associated treatments, and reasons for discontinuation of treatment. All the patients continued combination treatment with csDMARDS, immunosuppressants, and antifibrotic agents during follow-up.

The number of patients (%) who required hospitalization with a serious adverse event was not significantly higher compared to the rest of the patients (83.3% vs. 66.7%; *p* = 0.406), neither compared to patients with non-serious infection (83.3% vs. 43.9%; *p* = 0.478). There were also no significant differences in the dose of corticosteroids (mg/day) between patients were hospitalized with a serious adverse event compared to the rest of the patients (5.0 (2.5–10.0) vs. 3.75 (0.0–5.6); *p* = 0.175), neither compared to patients with non-severe infection (5.0 (2.5–7.5) vs. 5.0 (2.5–5.0); *p* = 0.526).

### 3.4. Factors Associated with Progression of Lung Disease in Patients with RA-ILD Treated with Abatacept

Appendix A shows the results of the bivariate analysis of patients with RA-ILD treated with abatacept whose lung disease did not progress and their associated baseline characteristics. Compared with patients whose condition improved/stabilized, those whose disease progressed/who died from ILD were characterized by greater inflammatory activity (mean (SD) DAS28-ESR, 4.4 (1.6) vs. 3.0 (1.2) mg/L; *p* = 0.012) and higher CRP (median (IQR), 20.0 (6.1–32.0) vs. 6.7 (5.0–13.1) mg/L; *p* = 0.015). In addition, more patients were taking corticosteroids (n (%), 25 (61.0) vs. 14 (87.5) mg/L; *p* = 0.041), had lower values at onset for FVC of ILD (mean (SD), 66.9 (23.9) vs. 81.0 (12.9) mg/L; *p* = 0.013) and DLCO-SB (mean (SD), 57.0 (15.5) vs. 72.7 (12.3) mg/L; *p* = 0.040), and less frequently took the combination of methotrexate and abatacept (n (%), 3 (18.8) vs. 19 (46.3.2) mg/L; *p* = 0.044).

Table 4 shows the results of the Cox multivariate analysis (dependent variable: progression or death) for 57 patients with RA-ILD over a median (IQR) time receiving abatacept of 27.3 (12.2–42.8) months. In 16/57 patients, the disease progressed, or the patient died. As shown in this model (Table 4), patients with greater inflammatory activity (DAS28-ESR) at initiation of abatacept were at a higher risk of progression and death, whereas higher FVC and DLCO-SB values at initiation of abatacept were associated with lower risk of progression and death.

## 4. Discussion

In the present study, we prospectively evaluated lung function and joint status in 57 patients with RA-ILD treated with abatacept. Lung disease stabilized or improved in 70% of patients, whereas it worsened, or the patient died, after a median of 27.3 months in a third of cases. These results are consistent with those observed in other retrospective cohorts with respect to the potential beneficial effect of abatacept in RA-ILD. During recent years, abatacept has been associated with improvement and stabilization of lung function in patients with RA-ILD in small case series [42,43,44]. Similarly, in their large-scale multicenter retrospective study of 263 patients with RA-ILD treated with abatacept, Fernández-Díaz et al. [26] found that disease had stabilized on HRCT in around 80% of patients and in the PFT in 90% after 12 months of follow-up. Stabilization of ILD with abatacept has been seen previously also in other condition such as in systemic sclerosis [45]. More recently, Tardella et al. [46] prospectively analyzed lung function in 44 patients with RA-ILD treated with abatacept for a median of 18 months and reported progression on HRCT in only 11% of cases. Compared with the abovementioned studies, data from the present study reveal a slight increase in the number of patients with RA-ILD whose disease progressed. This may be because of the somewhat longer follow-up period, the prospective nature of data collection, and the fact that when the patients initiated abatacept, they had a longer median course of ILD and poorer lung function values.

We observed that lung function worsened before therapy with abatacept than after in patients with RA. Following treatment, we also observed that disease had stabilized on HRCT in two-thirds of patients and that the distribution was equivalent for UIP and NSIP. Similarly, other authors have found an association between radiologic pattern in RA-ILD and effectiveness of abatacept [27,47,48]. Fernández-Díaz et al. [26] performed a subanalysis to compare the various radiologic patterns and did not find differences between the groups. Cassone et al. [47] observed a numerical improvement in patients with RA-ILD and the NSIP pattern treated with abatacept, although the differences were not significant.

Nevertheless, not all patients treated with abatacept progressed favorably. In 22.8% of cases the patient’s condition worsened, and 5.3% died. This unfavorable course was associated with factors such as poor control of joint inflammation (DAS28-ESR). Previous studies have shown that poorer control of joint involvement carries a higher risk of extra-articular manifestations, including ILD [49], and more pronounced progression of RA-ILD [50], whereas improvement in arthritis is associated with stabilization of lung disease [26]. Joint involvement and previous progression of lung disease improved at the end of follow-up in patients treated with abatacept, suggesting that abatacept could have a beneficial disease-modifying effect in these patients because of various mechanisms. On the one hand, it could have a direct effect on lung inflammation, since experimental studies have shown CTLA-4 to play a major role in lung inflammation in other lung diseases, such as hypersensitivity pneumonitis [26,51]. On the other, it could have an indirect effect by diminishing systemic inflammation in RA. This possibility has also been put forward by other authors with other disease-modifying drugs such as methotrexate. Rojas-Serrano et al. [11] and Juge et al. [52] showed that patients with ILD-RA treated with methotrexate not only have a greater risk of developing ILD, but that, in addition, methotrexate could improve survival by better controlling systemic inflammation in affected patients.

The multivariate analysis also showed that, in patients with RA-ILD treated with abatacept, the more pronounced impairment of FVC and DLCO at initiation of treatment was associated with poorer progression of lung disease. This finding has been reflected in various studies, thus pointing to an association with greater progression of lung disease and mortality [53,54]. However, less attention has been given to the analysis of factors associated with progression and mortality in patients with RA-ILD treated with abatacept. A recent systematic review attempted to synthesize available evidence on abatacept in patients with RA-ILD [55]. The authors reported efficacy and safety data from nine retrospective observational studies. Only the study by Cassone et al. [47] analyzed factors potentially associated with progression of lung disease. The authors evaluated 44 patients with RA-ILD treated with abatacept for ≥6 months. In 18% of those patients whose HRCT scan indicated progression, this tended to be associated with more pronounced impairment of FVC (*p* = 0.07). In relation to treatment, although corticosteroids were associated with progression of lung disease in the univariate analysis, it was finally not associated in the multivariate analysis. The sample size may not have been sufficient to show this association.

As for the safety profile of abatacept, we found the median survival to progression of lung disease to be 72.0 months. Although only 10.5% of patients were hospitalised with a serious adverse event, nearly half did develop infection over a relatively short period. These data are similar to those reported in a systematic review that evaluated safety of abatacept in patients with RA-ILD [55] and found that the drug was maintained in 76.4%, with only 10.6% of patients experiencing severe adverse effects. These data are also consistent with the findings of a meta-analysis showing that abatacept is associated with a lower risk of severe infection than other bDMARDs [56]. Therefore, it seems to have a more favourable safety profile that other biologics in patients with RA-ILD, because infections in this population are associated with high morbidity and mortality.

The present study is subject to a series of limitations. First, the sample size may be insufficient to reveal significant differences in some factors associated with progression. In this sense, a worsening of 20% in HRCT could be a severe progression and our study could underestimate this outcome. However, it is the only prospective study with such a high number of RA-ILD patients treated with abatacept, thus enabling us to provide a detailed description of progression of lung disease and joint disease at different times during the clinical course and even identify factors associated with progression in affected patients. Furthermore, we did not directly compare abatacept with other DMARDs, since, during this period, patients were selected for treatment with abatacept under conditions of daily clinical practice and our objective was not to perform a clinical trial. Nevertheless, we were able to perform direct comparisons with data from other studies and meta-analyses. On the other hand, the comparison between ILD diagnosis and baseline may be unbalanced by the patients with different duration of ILD. However, we have observed that lung progression or stabilization in patients with RA-ILD treated with abatacept was not associated with the duration of ILD. In our study there were six patients with more than 6 months of follow-up but less than 12 months. These six patients were excluded from the analysis at 12 months. However, our main objective has been the pulmonary evolution at the end of the follow-up, and we also have most of the patients with data at 12 months to study their evolution. Lastly, while the various studies that evaluate the effectiveness of abatacept used different criteria to evaluate progression of lung disease, thus hampering comparison, we evaluated progression using both PFT and HRCT and reported all data for each of these tests. Therefore, we were able to compare our findings with those of the other studies. In addition, we have evaluated pulmonary function and progression at the start of treatment with abatacept, at 12 months, and end of follow-up with abatacept. Although we have not evaluated the data 12 months before starting abatacept, we have considered that the data of pulmonary function and progression at initiation of treatment and during follow-up have shown the progression of ILD before and after treatment with abatacept.

## 5. Conclusions

In conclusion, around 70% of patients with RA-ILD treated with abatacept achieved stable lung function and improved control of inflammation after 27.3 months of follow-up, although lung disease progressed in one third. The factors associated with more marked progression of lung disease and mortality were greater inflammatory activity and low DLCO and FVC values. These factors should be considered at initiation of abatacept and during follow-up to identify patients at risk of progression and consider alternative treatments. Furthermore, abatacept seems to have a favorable safety profile in patients with RA-ILD. This favorable safety profile could be very important in affected patients, especially those who are vulnerable to infection.

## Figures and Tables

**Figure 1 biomedicines-10-01480-f001:**
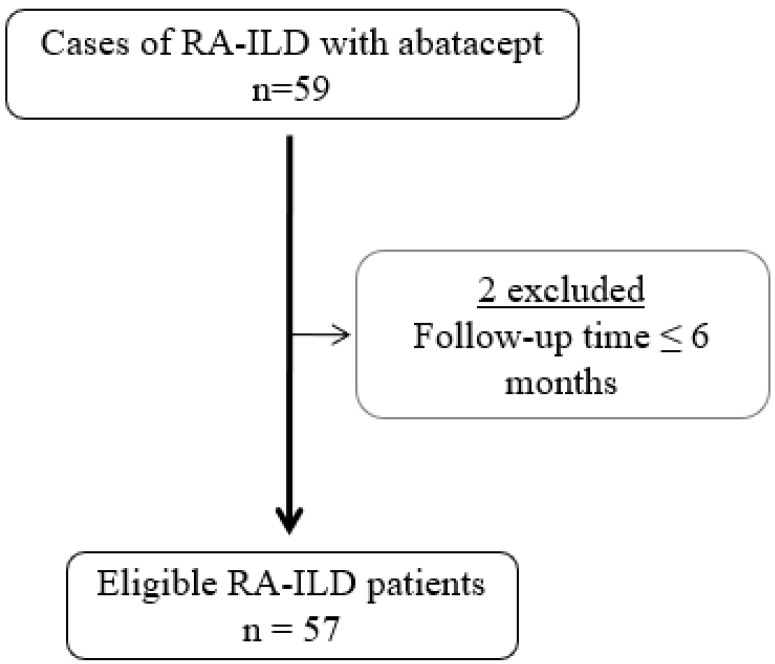
Patient flowchart.

**Figure 2 biomedicines-10-01480-f002:**
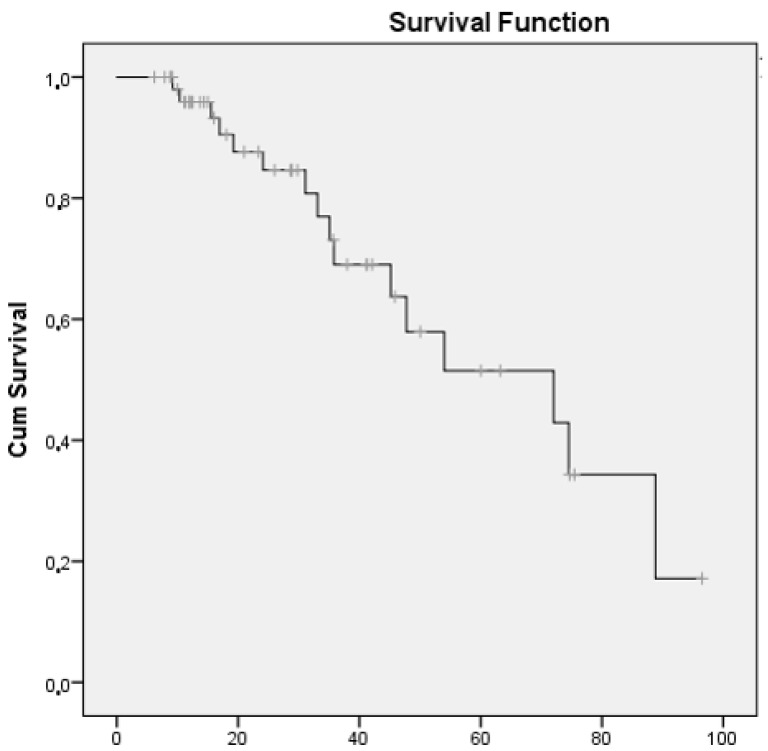
Survival curve for patients with RA-ILD receiving abatacept whose disease had progressed or who had died.

**Figure 3 biomedicines-10-01480-f003:**
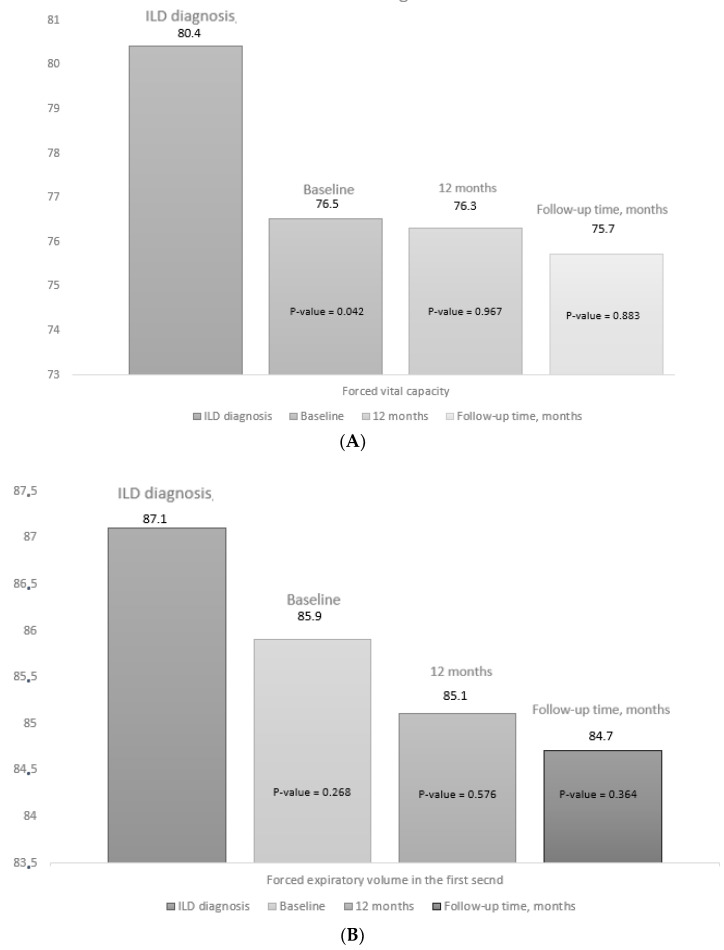
Progress of pulmonary function tests. Figure 3 shows the data for the pulmonary function tests at diagnosis of ILD, at the baseline visit (V0), at 12 months (V12), and at the final visit (FV): (**A**) forced vital capacity; (**B**) forced expiratory volume in the first second, (**C**) diffusing capacity of the lung for carbon monoxide. The *p* value for baseline (V0) is the result of comparing values between V0 and the date of diagnosis of ILD. The *p* value for 12 months (V12) and at the end of follow-up (FV) is the result of the comparison between V12 and FV and baseline (V0). The median (IQR) follow-up time was 27.3 (12.2–42.8) months.

**Table 1 biomedicines-10-01480-t001:** Baseline characteristics of 57 patients with RA-ILD treated with abatacept.

Variable	Sample = 57
Epidemiologic characteristics	
Female sex, n (%)	32 (56.1)
Caucasian race, n (%)	56 (98.2)
Age in years, mean (SD)	67.7 (22.0)
Clinical and laboratory characteristics	
Smoking status	
Nonsmoker, n (%)	39 (68.4)
Smoker, n (%)	9 (15.8)
Ex-smoker, n (%)	9 (15.8)
Body mass index, mean (SD)	27.9 (5.1)
Time since diagnosis of RA, months, median (IQR)	120.6 (66.0–220.9)
Diagnostic delay, months, median (IQR)	6.3 (3.7–16.0)
Time since diagnosis of ILD, months, median (IQR)	48.0 (25.8–93.9)
Positive rheumatoid factor (>10 U/mL), n (%)	54 (94.7)
Positive antipeptide citrullinated antibody (>20 U/mL), n (%)	48 (84.2)
Erosive disease, n (%)	36 (63.2)
Inflammatory activity	
28-joint Disease Activity Score, mean (SD)	3.8 (1.5)
Health Assessment Questionnaire, median (IQR)	1.1 (0.5–1.5)
C-reactive protein (mg/dl), median (IQR)	7.0 (2.5–20.7)
Erythrocyte sedimentation rate (mm/h), median (IQR)	33.0 (15.0–47.0)
Conventional synthetic DMARDs	47 (82.5)
Methotrexate, n (%)	22 (38.5)
Leflunomide, n (%)	17 (29.8)
Sulfasalazine, n (%)	2 (3.6)
Hydroxychloroquine, n (%)	6 (10.6)
Immunosuppressant	5 (8.8)
Mycophenolate, n (%)	3 (5.3)
Azathioprine, n (%)	2 (3.5)
Antifibrotic agents, nintedanib n (%)	1 (1.8)
Baseline corticosteroids, n (%)	39 (68.4)
Baseline dose of corticosteroids (grams), median (IQR)	5.0 (2.5–5.0)

Abbreviations: RA: rheumatoid arthritis; ILD: interstitial lung disease; DMARD: disease-modifying antirheumatic drug; SD: standard deviation; IQR: interquartile range.

**Table 2 biomedicines-10-01480-t002:** Progress of symptoms and lung disease at the end of follow-up in 57 patients with RA-ILD treated with abatacept.

Variable	Baseline	12 Months	End of Follow-Up	*p*-Value Baseline vs. 12 Months	*p*-Value Baseline vs. End of Follow-Up
Follow-up in months, median (RIC)	-	12.6 (11.9–12.8)	27.3 (12.2–42.8)	-	-
Lung function					
Oxygen saturation, mean (SD)	95.9 (2.4)	95.3 (3.6)	95.1 (3.9)	0.340	0.171
Pulmonary function tests					
FVC predicted (%), mean (SD)	76.5 (18.2)	76.3 (16.1)	75.7 (14.4)	0.990	0.883
FVC < 80%, n (%)	24 (42.1)	22 (43.1)	27 (47.3)	0.690	0.202
FVC ≥ 80%, n (%)	33 (57.9)	29 (56.8)	30 (52.6)	0.690	0.202
FEV_1_ predicted (%), mean (SD)	85.9 (10.4)	85.1 (19.1)	84.7 (13.9)	0.560	0.364
DLCO-SB predicted (%), mean (SD)	62.5 (16.5)	60.6 (16.0)	59.7 (15.7)	0.340	0.133
Clinical course					
Progression, n (%)	25 (43.8) *	10 (19.6)	13 (22.8)	0.014	0.036
Stabilization, n (%)	29 (50.8) *	34 (66.6)	38 (66.7)		
Improvement, n (%)	3 (5.3) *	7 (13.7)	6 (10.5)		
HRCT					
Radiological type				0.540	0.297
UIP, n (%)	34 (59.6)	31 (60.7)	36 (63.2)		
NSIP, n (%)	18 (31.6)	16 (31.3)	16 (28.1)		
Fibrotic NSIP, n (%)	5 (8.8)	4 (7.8)	5 (8.8)		
Clinical course				0.019	0.028
Progression, n (%)	20 (35.0) *	10 (19.6)	13 (22.8)		
Stabilization, n (%)	34 (59.6) *	35 (68.6)	39 (68.4)		
Improvement, n (%)	3 (5.3) *	6 (11.7)	5 (8.8)		
Overall progress of lung disease **				0.010	0.019
Improvement, n (%)	3 (5.3) *	7 (13.7)	6 (10.5)		
Stabilization, n (%)	28 (48.4) *	34 (66.6)	35 (61.4)		
Worsening, n (%)	26 (45.6) *	10 (17.5)	13 (22.8)		
Death, n (%)	-	-	3 (5.3)		
Inflammatory activity					
DAS28, mean (SD)	3.8 (1.5)	3.0 (1.6)	3.1 (1.2)	0.019	0.024
C-reactive protein (mg/dl), median (IQR)	7.0 (5.0–17.7)	3.6 (2.4–10.1)	3.3 (2.6–7.9)	0.041	0.018
ESR (mm/h), median (IQR)	33.0 (15.0–47.0)	19.0 (6.0–32.0)	20.0 (8.0–29.0)	0.045	0.046
HAQ, median (IQR)	1.0 (0.2–1.8)	1.0 (0.3–1.8)	1.1 (0.6–1.9)	0.629	0.424
Corticosteroids, n (%)	39 (68.4)	30 (58.8)	28 (48.4)	0.360	0.040

Abbreviations: RA: rheumatoid arthritis; ILD: interstitial lung disease; DMARD: disease-modifying antirheumatic drug; SD: standard deviation; IQR: interquartile range; FVC: forced vital capacity; FEV_1_: forced expiratory volume in the first second; DLCO: diffusing capacity of the lung for carbon monoxide; UIP: usual interstitial pneumonia; NSIP: nonspecific interstitial pneumonia; HRCT: high-resolution computed tomography; DAS28: 28-joint Disease Activity Score; CRP: C-reactive protein; ESR: erythrocyte sedimentation rate; HAQ: Health Assessment Questionnaire; * Data from baseline vs. date of diagnosis of ILD; ** Progression of lung disease overall taking into account HRCT and pulmonary function testing (FCV and DLCO). Pulmonary function tests considered progression (decrease in FVC > 10% or in DLCO > 15), stabilization (no progression or increase in FVC < 10% or in DLCO < 15%) and improvement (FVC ≥ 10% or DLCO ≥ 15%). HRCT considered progression (extension ≥ 20%), stabilization (extension < 20%) and improvement (decrease in extension).

**Table 3 biomedicines-10-01480-t003:** Adverse events in 57 patients with RA-ILD treated with abatacept.

Variable	Sample = 57
Infections, n (%)	25 (43.9)
Respiratory infection, n (%)	20 (35.0)
Other infections, n (%)	6 (10.5)
Cutaneous disease, n (%)	2 (3.5)
Urinary infection, n (%)	4 (7.0)
Transient neutropenia, n (%)	1 (3.5)
Hospitalization, n (%)	6 (10.5)
Reasons for hospitalization	
Progression of ILD, n (%)	3 (5.2)
Respiratory infection, n (%)	2 (3.5)
Pyelonephritis, n (%)	1 (3.5)
Mortality, n (%)	3 (5.2)

Abbreviations: RA: rheumatoid arthritis; ILD: interstitial lung disease.

**Table 4 biomedicines-10-01480-t004:** Multivariate analysis of factors associated with progression of lung disease and mortality in patients with RA-ILD at initiation of treatment with abatacept. Cox regression model (adjusted for time receiving abatacept).

Variable	Univariate HR (95%CI)	Multivariate HR (IC 95%)	*p*-Value
Age, years	1.000 (0.974–1.028)		
Male sex	2.975 (0.618–7.109)		
Current or previous smoking history	1.292 (0.386–4.321)		
ACPA (>20 U/mL)	0.417 (0.096–1.809)		
UIP radiological pattern	4.480 (0.896–12.390)		
DAS28-ESR	2.001 (1.194–3.354)	2.520 (1.038–3.120)	0.041
FVC predicted (%)	0.879 (0.819–0.944)	0.826 (0.704–0.969)	0.019
DLCO-SB predicted (%)	0.889 (0.835–0.946)	0.837 (0.723–0.969)	0.018
Combination with methotrexate	0.267 (0.066–0.996)		
Corticosteroids (grams)	1.272 (1.047–1.545)		

Abbreviations: RA: rheumatoid arthritis; ILD: interstitial lung disease; ACPA: anticitrullinated peptide antibody; UIP: usual interstitial pneumonia; DAS28-ESR: 28-joint Disease Activity Score with erythrocyte sedimentation rate; FVC: forced vital capacity; DLCO-SB: diffusing capacity of the lung for carbon monoxide; Independent variables included in the equation at V0: sex, age, FVC, DAS28-ESR, DLCO-SB, methotrexate in combination, corticosteroids. Nagelkerke R^2^: 0.47.

## Data Availability

Data presented in this study are available on request from the corresponding author.

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
