# Peer review of "Safety and Effectiveness of Abatacept in a Prospective Cohort of Patients with Rheumatoid Arthritis–Associated Interstitial Lung Disease"

_biomedicines, 2022, doi:10.3390/biomedicines10071480_

Round 1

Reviewer 1 Report

The work seems to me well conducted with adequate bibliography. The treated topic is very interesting and the results of the surdio are very useful for the rheumatologist community, who will be able to treat patients with rheumatoid arthritis and pulmonary interstitial disease.

Author Response

Comments for the reviewers

We would like to thank the editor for considering our work for publication in “Biomedicines” and the reviewers for their comments, which have helped to improve the quality of our manuscript.

Below, we provide a point-by-point reply to the comments.

Reviewer #1: 

  1. Comments and Suggestions for Authors

The work seems to me well conducted with adequate bibliography. The treated topic is very interesting and the results of the surdio are very useful for the rheumatologist community, who will be able to treat patients with rheumatoid arthritis and pulmonary interstitial disease.

Reply: We appreciate your comment.

Reviewer #2: 

The authors describe the effects of abatacept in 57 patients with RA-ILD over a median of 27 months and report in detail on the influence of this agent on the associated ILD using well-defined criteria. They compare their results to others in the literature. The study is well conducted but I have a few comments:

  1. Within the definitions of improved vs stablised vs deteriorated, how did the authors classify those whose lung function was stable but HRCT deteriorated and vice versa? The description given does not clarify this.

Reply: In our study we have considered worsening or progression if the patient presented any of the following criteria: forced vital capacity (FVC) decrease ≥ 10%, diffusion capacity for carbon monoxide (DLCO) decrease ≥ 15%, radiological increase of lung fibrotic signs or death. We agree with the reviewer that the definition as formulated may be confusing. Based on reviewer comments, we have clarified the definition. We appreciate your comment and have modified this in the manuscript.

-Pag. 3; line 114-120: Working definitions and variables: “The main variable was the effectiveness of abatacept according to the outcome of ILD at the end of follow-up (FV) with respect to the following: (1) improvement (i.e., improvement in forced vital capacity [FVC] ≥10% or in the diffusing capacity of the lung for carbon monoxide [DLCO] ≥15% and no radiological progression); (2) non progression (stabilization or improvement in FVC <10% or in DLCO <15% and no radiological progression); (3) progression (worsening of FVC >10% or DLCO >15% or radiological progression); and (4) death. “

  1. Although'only'6 (10%) of patients were hospitalised with a serious adverse event, nearly half did develop infection over a relatively short period. Although this is possibly less than for other biologics, the authors do not comment on the potential influence of steroids in these patients. Were those hospitalised with infection on higher steroid doses?

Reply: Following the recommendations of the reviewer, we have analyzed the treatment with glucocorticoids in the patients of our cohort who required hospitalization during follow-up and we found no significant differences compared to the rest in the number of patients treated with glucocorticoids or in terms of dose. Likewise, we also found no differences between patients with serious and non-serious infections. We have inserted the following results into the text:

-Pag. 10; line 268-274: “The number of patients (%) who required hospitalization with a serious adverse event was not significantly higher compared to the rest of the patients (83.3% vs 66.7%; p=0.406), neither compared to patients with non-serious infection (83.3% vs 43.9%; p=0.478). There were also no significant differences in the dose of corticosteroids (mg/day) between patients were hospitalized with a serious adverse event compared to the rest of the patients (5.0 [2.5-10.0] vs 3.75 [0.0-5.6]; p=0.175), neither compared to patients with non-severe infection (5.0 [2.5-7.5] vs 5.0 [2.5-5.0]; p=0.526).”

-Pag 12; line 367-376: “As for the safety profile of abatacept, we found the median survival to progression of lung disease to be 72.0 months. Although only 10.5% of patients were hospitalised with a serious adverse event, nearly half did develop infection over a relatively short period. These data are similar to those reported in a systematic review that evaluated safety of abatacept in patients with RA-ILD (55) and found that the drug was maintained in 76.4%, with only 10.6% of patients experiencing severe adverse effects. These data are also consistent with the findings of a meta-analysis showing that abatacept is associated with a lower risk of severe infection than other bDMARDs (56). Therefore, it seems to have a more favourable safety profile that other biologics in patients with RA-ILD, because infections in this population are associated with high morbidity and mortality.”

  1. Among those 46% of patients whose disease had previously progressed prior to abatacept, was the outcome worse than those whose disease had been previously stable?  Although this may be inferred from the relationships between worse baseline VC / TLco and progression during therapy, it would be helpful to see this specifically addressed

Reply:  Thanks for the comment. Among the 26 patients who had progressed before being treated with abatacept, half progressed and the other half did not, while among the patients whose disease had been stable before abatacept, virtually all remained stable throughout the follow-up period. We have made the following changes to the results:

-Pag. 6; line 214-217: “Among the 26 (46%) patients who had previously progressed prior to abatacept, 13/26 (50%) had progression at the end of follow-up and another 13/26 (50%) had no progression (p=1.000), whereas among patients whose disease had been stable before abatacept, 3/31 (9.7%) progressed and N28/ 31 (90.3%) did not progress (p=0.001).”

  1. A minor point: I find the sentence on Page 6 from line 203-205 somewhat confusing. Can this be clarified please.

Reply: According to the reviewer, we have clarified this sentence:

-Pag 6; line 205-207: “In addition, as can be seen in Table 2, the progression of lung disease prior to starting abatacept was more marked than during the period of treatment with abatacept (45.6% vs 22.8%; p=0.019).”

Reviewer #3: 

Conclusions clearly presented. The study can be surely used for further extended researches

Reply: We appreciate your comment.

Reviewer #4: 

The manuscript presents the results of a small cohort study of RA-ILD patients treated with Abatacept. The nature of the study presents with several limitations, which should be better highlighted and do now allow to draw real conclusions about a possible efficacy of abatacept.

In addition, I would consider multiple major methodological limitation, which must be addressed to improve the overall quality of the manuscript:

  1. FV being in 2021 would be the same timepoint for patients enrolled in 2020; I do not see the real benefit of this timepoint overall, as 25% of patients have a FU duration below 12.2 years.

Reply:  In line with the reviewer's suggestion, we thought it might be interesting to see the evolution of the patients at 12 months and at the end of follow-up. There were only 6 patients with > 6 months of follow-up and less than 12 months. These 6 patients were excluded from the analysis at 12 months. But most of the patients had evolution at 12 months. So, we believe that this could be interesting to see the evolution of the majority of the sample. We have added this comment in limitations.

-Pag. 12; line 391-395: “In our study there were 6 patients with more 6 months of follow-up but less than 12 months. These 6 patients were excluded from the analysis at 12 months. However, our main objective has been the pulmonary evolution at the end of the follow-up, and we also have most of the patients with data at 12 months to study their evolution.”

  1. the comparison between ILD diagnosis and Baseline for all parameters is not really informative, given the very wide range of ILD duration. The comparison does not take into account the time between the two assessments and, therefore, may be strongly unbalanced by the patients with twice longer disease duration.

Reply: As the reviewer has commented, the comparison between ILD diagnosis and baseline may be strongly unbalanced by the patients with twice longer disease duration. However, as we can see in Supplementary table 5 (Factors associated with progression of lung disease in patients with RA-ILD treated with abatacept.), lung progression or stabilization in patients with RA-ILD treated with abatacept was not associated with the duration of ILD. We have also added this comment in limitations of the study.

- Pag. 14; line 432: Supplementary table 5: Factors associated with progression of lung disease in patients with RA-ILD treated with abatacept.

Variable

Improvement/stabilization (n=41)

Progression/death (n=16)

p-value

Epidemiological characteristics

   Sex, female, n (%)

21 (51.2)

11 (68.8)

0.231

   Age, years, mean (SD)

67.6 (14.9)

67.8 (11.6)

0.970

   Caucasian race, n (%)

41 (100.0)

16 (100.0)

1.000

Clinical and laboratory characteristics

   Smoking

0.677

      Never smoked, n (%)

28 (68.3)

10 (62.5)

      Smoked, n (%)

13 (31.7)

6 (37.5)

   Duration of RA, months, median (IQR)

98.3 (63.7-230.9)

139.0 (70.6-200.9)

0.750

   Duration of ILD, months, median (IQR)

45.0 (17.7-93.1)

53.5 (39.3-95.8)

0.390

   Positive RF (>10 U/ml), n (%)

40 (97.6)

14 (87.5)

0.187

   ACPA (>20 U/ml), n (%)

36 (87.8)

12 (75.0)

0.234

Inflammatory activity

      DAS 28-ESR, mean (SD)

3.0 (1.2)

      4.4 (1.6)

       0.012

      HAQ, median (IQR)

0.8 (0.2-1.6)

       1.0 (1.0-1.3)

       0.503

      C-reactive protein (mg/dl), median (IQR)

6.7 (5.0-13.1)

       20.0 (6.1-32.0)

       0.015

      ESR (mm/h), median (IQR)

24.5 (11.7-41.5)

     36.0 (15.0-55.0)

       0.350

Radiological pattern

0.435

   UIP, n (%)

28 (68.3)

8 (50.0)

   NSIP, n (%)

10 (24.4)

6 (37.5)

   Fibrotic NSIP, n (%)

3 (7.3)

2 (12.5)

Baseline PFT

   FVC predicted (%), mean (SD)

81.0 (12.9)

66.9 (23.9)

0.013

   FEV1 predicted (%), mean (SD)

84.0 (12.6)

75.1 (28.2)

0.145

   DLCO-SB predicted (%), mean (SD)

72.7 (12.3)

57.0 (15.5)

0.040

Treatment

  Time to initiation of abatacept, median (IQR)

22.1 (12.0-41.2)

34.0 (17.5-52.4)

0.109

   Time receiving abatacept, median (IQR)

32.0 (17.5-42.4)

24.6 (10.7-40.8)

0.103

   Combined with a DMARD, n (%)

32 (78.0)

13 (81.3)

0.790

   Methotrexate, n (%)

19 (46.3)

3 (18.8)

0.044

   Leflunomide, n (%)

12 (30.0)

5 (31.3)

0.927

   Sulfasalazine, n (%)

1 (2.6)

1 (6.3)

0.507

   Hydroxychloroquine, n (%)

3 (7.3)

3 (18.3)

0.406

   Combined with immunosuppressants, n (%)

4 (10.0)

1 (6.3)

0.081

   Mycophenolate, n (%)

2 (5.2)

1 (6.3)

0.890

   Azathioprine, n (%)

2 (5.2)

0 (0.0)

0.570

   Corticosteroids, n (%)

25 (61.0)

14 (87.5)

0.041

Dose of corticosteroids (grams), median (IQR)

2.5 (0.0-5.0)

5.0 (5.0-10.0)

0.037

Abbreviations. RA: rheumatoid arthritis; ILD: interstitial lung disease; SD: standard deviation; IQR: interquartile range; RF: rheumatoid factor; ACPA: anticitrullinated peptide antibody; DAS28: 28-joint Disease Activity Score; ESR: erythrocyte sedimentation rate; HAQ: Health Assessment Questionnaire; UIP: usual interstitial pneumonia; NSIP: nonspecific interstitial pneumonia; PFT: pulmonary function testing; FVC: forced vital capacity; FEV1: forced expiratory volume in the first second; DLCO: diffusing capacity of the lung for carbon monoxide; SB: single breath; ABT: abatacept; DMARD: disease-modifying antirheumatic drug

-Pag. 12; line 388-391: “On the other hand, the comparison between ILD diagnosis and baseline may be unbalanced by the patients with different duration of ILD. However, we have observed that lung progression or stabilization in patients with RA-ILD treated with abatacept was not associated with the duration of ILD.”

  1. the 12 months evaluation is the real timepoint to consider in the study; what about comparing the change in FVC-DLCO between v0-v1 versus the change in the same between 12 months before starting abatacept and v0? That would show signal for different effect of abatacept and allow some comparison versus non-abatacept treatment.

Reply: The main variable of our study was the effectiveness of abatacept according to the outcome of ILD at the end of follow-up (FV) vs baseline, not the evaluation at 12 months. The secondary variables, we evaluated the change in FVC and DLCO, and radiologic progression at 12 months of treatment with abatacept (V12) and at the end of follow-up (FV) vs baseline. This show us about how the patients have been able to progress with the treatment. On the other hand, we have collected the data at the diagnosis of ILD compared to baseline allows us to see how the evolution of the patients has been before the initiation of treatment. Therefore, we consider that these points have shown our objectives. However, we have added in our limitations that we do not have the data 12 months before starting abatacept.

Table 2. Progress of symptoms and lung disease at the end of follow-up in 57 patients with RA-ILD treated with abatacept.

Variable

Baseline

12 months

End of follow-up

p-value

Baseline vs 12 months

p-value Baseline vs End of follow-up

Follow-up in months, median (RIC)

-

12.6 (11.9-12.8)

27.3 (12.2-42.8)

-

-

Lung function

Oxygen saturation, mean (SD)

95.9 (2.4)

95.3 (3.6)

95.1 (3.9)

0.340

0.171

Pulmonary function tests

FVC predicted (%), mean (SD)

76.5 (18.2)

76.3 (16.1)

75.7 (14.4)

0.990

0.883

FVC <80%, n (%)

24 (42.1)

22 (43.1)

27 (47.3)

0.690

0.202

FVC ≥80%, n (%)

33 (57.9)

29 (56.8)

30 (52.6)

0.690

0.202

FEV1 predicted (%), mean (SD)

85.9 (10.4)

85.1 (19.1)

84.7 (13.9)

0.560

0.364

DLCO-SB predicted (%), mean (SD)

62.5 (16.5)

60.6 (16.0)

59.7 (15.7)

0.340

0.133

Clinical course

Progression, n (%)

25 (43.8) *

10 (19.6)

13 (22.8)

0.014

0.036

Stabilization, n (%)

29 (50.8) *

34 (66.6)

38 (66.7)

Improvement, n (%)

3 (5.3) *

7 (13.7)

6 (10.5)

HRCT

Radiological type

0.540

0.297

UIP, n (%)

34 (59.6)

31 (60.7)

36 (63.2)

NSIP, n (%)

18 (31.6)

16 (31.3)

16 (28.1)

Fibrotic NSIP, n (%)

5 (8.8)

4 (7.8)

5 (8.8)

Clinical course

0.019

0.028

Progression, n (%)

20 (35.0) *

10 (19.6)

13 (22.8)

Stabilization, n (%)

34 (59.6) *

35 (68.6)

39 (68.4)

Improvement, n (%)

3 (5.3) *

6 (11.7)

5 (8.8)

Overall progress of lung disease**

0.010

0.019

Improvement, n (%)

3 (5.3) *

7 (13.7)

6 (10.5)

Stabilization, n (%)

28 (48.4) *

34 (66.6)

35 (61.4)

Worsening, n (%)

26 (45.6) *

10 (17.5)

13 (22.8)

Death, n (%)

-

-

3 (5.3)

Inflammatory activity

DAS28, mean (SD)

3.8 (1.5)

3.0 (1.6)

3.1 (1.2)

0.019

0.024

C-reactive protein (mg/dl), median (IQR)

7.0 (5.0-17.7)

3.6 (2.4-10.1)

3.3 (2.6-7.9)

0.041

0.018

ESR (mm/h), median (IQR)

33.0 (15.0-47.0)

19.0 (6.0-32.0)

20.0 (8.0-29.0)

0.045

0.046

HAQ, median (IQR)

1.0 (0.2-1.8)

1.0 (0.3-1.8)

1.1 (0.6-1.9)

0.629

0.424

Corticosteroids, n (%)

39 (68.4)

30 (58.8)

28 (48.4)

0.360

0.040

Abbreviations: RA: rheumatoid arthritis; ILD: interstitial lung disease; DMARD: disease-modifying antirheumatic drug; SD: standard deviation; IQR: interquartile range; FVC: forced vital capacity; FEV1: forced expiratory volume in the first second; DLCO: diffusing capacity of the lung for carbon monoxide; UIP: usual interstitial pneumonia; NSIP: nonspecific interstitial pneumonia; HRCT: high-resolution computed tomography; DAS28: 28-joint Disease Activity Score; CRP: C-reactive protein; ESR: erythrocyte sedimentation rate; HAQ: Health Assessment Questionnaire; * Data from baseline vs date of diagnosis of ILD; **Progression of lung disease overall taking into account HRCT and pulmonary function testing (FCV and DLCO). Pulmonary function tests considered progression (decrease in FVC >10% or in DLCO >15), stabilization (no progression or increase in FVC <10% or in DLCO <15%) and improvement (FVC ≥10% or DLCO ≥15%). HRCT considered progression (extension ≥ 20%), stabilization (extension < 20%) and improvement (decrease in extension).

-Pag 13; line 399-404: “In addition, we have evaluated pulmonary function and progression at the start of treatment with abatacept, at 12 months, and end of follow-up with abatacept. Although we have not evaluated the data 12 months before starting abatacept, we have considered that the data of pulmonary function and progression at initiation of treatment and during follow-up have shown the progression of ILD before and after treatment with abatacept.”

  1. the combined outcomes proposed are very weak. In particular, where were the cut-offs derived from a patients who se FVC il 8% higher has not improved? Combining Dlco decline and HRCT progression? How was ILD progression on HRCT determined? According to which visual scoring assessment? Increase in 20% is absolute or relative? A 20% absolute increase in ILD would represent a severe progressor and I would not expect the study to be able to pick up many events; this also implies that a 19% increase in ILD extent is not a radiologic progression? Hard to believe. No data at all is presented urvival ILD extent anyway, both at baseline and FU à I would suggest considering changes in FVC alone, with values somehow close to the MCID proposed for IPF (du Bois RM, Weycker D, Albera C, et al. Forced vital capacity in patients with urvivale pulmonary fibrosis: test properties and minimal clinically important urvivale. Am J Respir Crit Care Med 2011; 184: 1382–1389) otherwise take it from another CTD-ILD (10.1164/rccm.201709-1845OC). Otherwise, use the Goh et al 2017 definition of progression combining FVC and DLCO changes. 

Reply: We have used a combined variable that includes PFR and HRCT factors. However, we have also separately measured progression by HRCT and deterioration of CFV, DLCO as a variable of pulmonary progression. In fact, we think that this is a strength of our study. While the various studies that evaluate the effectiveness of abatacept used different criteria to evaluate progression of lung disease, thus hampering comparison, we evaluated progression using both PFT and HRCT and reported all data for each of these tests. Therefore, we were able to compare our findings with those of the other studies. In this sense, and following the reviewer's comments, we show the total progression and also by HRCT, FVC and DLCO, separately.

We agree with the reviewer, and we have evaluated progression in lung function test.

Regarding the progression of lung function test, we have considered progression decrease in forced vital capacity [FVC] >10% or in the diffusing capacity of the lung for carbon monoxide [DLCO] >15, stabilization as no progression or increase in FVC <10% or in DLCO <15% and improvement FVC ≥10% or DLCO ≥15%. In the evaluation by HRCT, we have considered progression with extension of 20% or more, stabilization with extension less than 20%, and improvement as decrease in extension. This is consistent with other articles described in the literature in patients with RA-ILD. This allows us to compare results and standardize this survival. We have added these results at the request of the reviewer. We have also added in limitations that a worsening of 20% in HRCT could be a severe progression and our study could underestimate this outcome.

- Song JW, Lee HK, Lee CK, Chae EJ, Jang SJ, Colby TV, et al. Clinical course and outcome of rheumatoid arthritis-related usual interstitial pneumonia. Sarcoidosis Vasc Diffuse Lung Dis 2013;30:103-12.

- Hyldgaard C, Ellingsen T, Hilberg O, Bendstrup E. Rheumatoid arthritis-associated interstitial lung disease: clinical characteristics and predictors of mortality. Respiration 2019;98:455-60.

-Kelly CA, Saravanan V, Nisar M, Arthanari S, Woodhead FA, Price-Forbes AN, et al; British Rheumatoid Interstitial Lung (BRILL) Network. Rheumatoid arthritis-related interstitial lung disease: associations, prognostic factors and physiological and radiological characteristics—a large multicentre UK study.  Rheumatology (Oxford) 2014;53:1676-82.

-Ito Y, Arita M, Kumagai S, Takei R, Noyama M, Tokioka F, et al. Radiological fibrosis score is strongly associated with worse survival in rheumatoid arthritis-related interstitial lung disease. Mod Rheumatol 2019;29:98-104.

-Jacob J, Hirani N, van Moorsel CHM, Rajagopalan S, Murchison JT, van Es HW, et al. Predicting outcomes in rheumatoid arthritis related interstitial lung disease. Eur Respir J. 2019;53:1800869.

-Pag 3; line 120-129, variables: “Similarly, as secondary variables, we evaluated the change in FVC and DLCO, and radiologic progression at 12 months of treatment with abatacept (V12) and at the end of follow-up (FV). The lung function was considered progression (decrease in forced vital capacity [FVC] >10% or in the diffusing capacity of the lung for carbon monoxide [DLCO] >15), stabilization (no progression or increase in FVC <10% or in DLCO <15%) and improvement (FVC ≥10% or DLCO ≥15%) (31, 32). Radiologic evaluation was considered progression (≥20% increase in the presence and extension of ground-glass opacities, reticulation, honeycombing, diminished attenuation, centrilobular nodules, other nodules, emphysema, and consolidation), stabilization (extension <20%) and improvement (decrease in extension) (33-35)”

-Pag 6; line 211-213: “At the end of follow-up, radiologic progression on HRCT was observed in 13/57 patients (22.8%), function pulmonary progression for Pulmonary function tests in 13/57 patients (22.8%), and 3/57 patients (5.3%) had died.”

-Pag 12; line 379-380: The present study is subject to a series of limitations. First, the sample size may be insufficient to reveal significant differences in some factors associated with progression. In this sense, a worsening of 20% in HRCT could be a severe progression and our study could underestimate this outcome. However, it is the only prospective study with such a high number of RA-ILD patients treated with abatacept, thus enabling us to provide a detailed description of progression of lung disease and joint disease at different times during the clinical course and even identify factors associated with progression in affected patients.

Table 2. Progress of symptoms and lung disease at the end of follow-up in 57 patients with RA-ILD treated with abatacept.

Variable

Baseline

12 months

End of follow-up

p-value

Baseline vs 12 months

p-value Baseline vs End of follow-up

Follow-up in months, median (RIC)

-

12.6 (11.9-12.8)

27.3 (12.2-42.8)

-

-

Lung function

Oxygen saturation, mean (SD)

95.9 (2.4)

95.3 (3.6)

95.1 (3.9)

0.340

0.171

Pulmonary function tests

FVC predicted (%), mean (SD)

76.5 (18.2)

76.3 (16.1)

75.7 (14.4)

0.990

0.883

FVC <80%, n (%)

24 (42.1)

22 (43.1)

27 (47.3)

0.690

0.202

FVC ≥80%, n (%)

33 (57.9)

29 (56.8)

30 (52.6)

0.690

0.202

FEV1 predicted (%), mean (SD)

85.9 (10.4)

85.1 (19.1)

84.7 (13.9)

0.560

0.364

DLCO-SB predicted (%), mean (SD)

62.5 (16.5)

60.6 (16.0)

59.7 (15.7)

0.340

0.133

Clinical course

Progression, n (%)

25 (43.8) *

10 (19.6)

13 (22.8)

0.014

0.036

Stabilization, n (%)

29 (50.8) *

34 (66.6)

38 (66.7)

Improvement, n (%)

3 (5.3) *

7 (13.7)

6 (10.5)

HRCT

Radiological type

0.540

0.297

UIP, n (%)

34 (59.6)

31 (60.7)

36 (63.2)

NSIP, n (%)

18 (31.6)

16 (31.3)

16 (28.1)

Fibrotic NSIP, n (%)

5 (8.8)

4 (7.8)

5 (8.8)

Clinical course

0.019

0.028

Progression, n (%)

20 (35.0) *

10 (19.6)

13 (22.8)

Stabilization, n (%)

34 (59.6) *

35 (68.6)

39 (68.4)

Improvement, n (%)

3 (5.3) *

6 (11.7)

5 (8.8)

Overall progress of lung disease**

0.010

0.019

Improvement, n (%)

3 (5.3) *

7 (13.7)

6 (10.5)

Stabilization, n (%)

28 (48.4) *

34 (66.6)

35 (61.4)

Worsening, n (%)

26 (45.6) *

10 (17.5)

13 (22.8)

Death, n (%)

-

-

3 (5.3)

Inflammatory activity

DAS28, mean (SD)

3.8 (1.5)

3.0 (1.6)

3.1 (1.2)

0.019

0.024

C-reactive protein (mg/dl), median (IQR)

7.0 (5.0-17.7)

3.6 (2.4-10.1)

3.3 (2.6-7.9)

0.041

0.018

ESR (mm/h), median (IQR)

33.0 (15.0-47.0)

19.0 (6.0-32.0)

20.0 (8.0-29.0)

0.045

0.046

HAQ, median (IQR)

1.0 (0.2-1.8)

1.0 (0.3-1.8)

1.1 (0.6-1.9)

0.629

0.424

Corticosteroids, n (%)

39 (68.4)

30 (58.8)

28 (48.4)

0.360

0.040

Abbreviations: RA: rheumatoid arthritis; ILD: interstitial lung disease; DMARD: disease-modifying antirheumatic drug; SD: standard deviation; IQR: interquartile range; FVC: forced vital capacity; FEV1: forced expiratory volume in the first second; DLCO: diffusing capacity of the lung for carbon monoxide; UIP: usual interstitial pneumonia; NSIP: nonspecific interstitial pneumonia; HRCT: high-resolution computed tomography; DAS28: 28-joint Disease Activity Score; CRP: C-reactive protein; ESR: erythrocyte sedimentation rate; HAQ: Health Assessment Questionnaire; * Data from baseline vs date of diagnosis of ILD; **Progression of lung disease overall taking into account HRCT and pulmonary function testing (FCV and DLCO). Pulmonary function tests considered progression (decrease in FVC >10% or in DLCO >15), stabilization (no progression or increase in FVC <10% or in DLCO <15%) and improvement (FVC ≥10% or DLCO ≥15%). HRCT considered progression (extension ≥ 20%), stabilization (extension < 20%) and improvement (decrease in extension).

-Pag 17, references:

  1. Hyldgaard C, Ellingsen T, Hilberg O, Bendstrup E. Rheumatoid Arthritis-Associated Interstitial Lung Disease: Clinical Characteristics and Predictors of Mortality. Respiration. 2019;98(5):455-60.
  2. Song JW, Lee HK, Lee CK, Chae EJ, Jang SJ, Colby TV, et al. Clinical course and outcome of rheumatoid arthritis-related usual interstitial pneumonia. Sarcoidosis Vasc Diffuse Lung Dis. 2013;30(2):103-12.
  3. Kelly CA ea. Rheumatoid arthritis-related interstitial lung disease: associations, prognostic factors and physiological and radiological characteristics. 2015.
  4. Ito Y, Arita M, Kumagai S, Takei R, Noyama M, Tokioka F, et al. Radiological fibrosis score is strongly associated with worse survival in rheumatoid arthritis-related interstitial lung disease. Mod Rheumatol. 2019;29(1):98-104.
  5. Jacob J, Hirani N, van Moorsel CHM, Rajagopalan S, Murchison JT, van Es HW, et al. Predicting outcomes in rheumatoid arthritis related interstitial lung disease. Eur Respir J. 2019;53(1).

  1. table 2 should present also all data at 12months, as it is one of the study timepoints. The measurece between “clinical course” and “overall clinical course” is very measure, as not specified in the methods section. Is the first one only based on PFT?

Reply: In line with the reviewer's suggestion, we have modified table 2 and we have added the data at 12 months of follow-up. The main variable was the effectiveness of abatacept according to the outcome of ILD at the end of follow-up (FV) with respect to the following: (1) improvement (i.e., improvement in forced vital capacity [FVC] ≥10% or in the diffusing capacity of the lung for carbon monoxide [DLCO] ≥15% and no radiological progression); (2) non-progression (stabilization or improvement in FVC <10% or in DLCO <15% and no radiological progression); (3) progression (worsening of FVC >10% or DLCO >15% or radiological progression); and (4) death. Similarly, as secondary variables, we evaluated the change in FVC and DLCO, and radiologic progression at 12 months of treatment with abatacept (V12) and at the end of follow-up (FV). The lung function was considered progression (decrease in forced vital capacity [FVC] >10% or in the diffusing capacity of the lung for carbon monoxide [DLCO] >15), stabilization (no progression or increase in FVC <10% or in DLCO <15%) and improvement (FVC ≥10% or DLCO ≥15%). Radiologic evaluation was considered progression (≥20% increase in the presence and extension of ground-glass opacities, reticulation, honeycombing, diminished attenuation, centrilobular nodules, other nodules, emphysema, and consolidation), stabilization (extension <20%) and improvement (decrease in extension). We have clarified this in manuscript.

Table 2. Progress of symptoms and lung disease at the end of follow-up in 57 patients with RA-ILD treated with abatacept.

Variable

Baseline

12 months

End of follow-up

p-value

Baseline vs 12 months

p-value Baseline vs End of follow-up

Follow-up in months, median (RIC)

-

12.6 (11.9-12.8)

27.3 (12.2-42.8)

-

-

Lung function

Oxygen saturation, mean (SD)

95.9 (2.4)

95.3 (3.6)

95.1 (3.9)

0.340

0.171

Pulmonary function tests

FVC predicted (%), mean (SD)

76.5 (18.2)

76.3 (16.1)

75.7 (14.4)

0.990

0.883

FVC <80%, n (%)

24 (42.1)

22 (43.1)

27 (47.3)

0.690

0.202

FVC ≥80%, n (%)

33 (57.9)

29 (56.8)

30 (52.6)

0.690

0.202

FEV1 predicted (%), mean (SD)

85.9 (10.4)

85.1 (19.1)

84.7 (13.9)

0.560

0.364

DLCO-SB predicted (%), mean (SD)

62.5 (16.5)

60.6 (16.0)

59.7 (15.7)

0.340

0.133

Clinical course

Progression, n (%)

25 (43.8) *

10 (19.6)

13 (22.8)

0.014

0.036

Stabilization, n (%)

29 (50.8) *

34 (66.6)

38 (66.7)

Improvement, n (%)

3 (5.3) *

7 (13.7)

6 (10.5)

HRCT

Radiological type

0.540

0.297

UIP, n (%)

34 (59.6)

31 (60.7)

36 (63.2)

NSIP, n (%)

18 (31.6)

16 (31.3)

16 (28.1)

Fibrotic NSIP, n (%)

5 (8.8)

4 (7.8)

5 (8.8)

Clinical course

0.019

0.028

Progression, n (%)

20 (35.0) *

10 (19.6)

13 (22.8)

Stabilization, n (%)

34 (59.6) *

35 (68.6)

39 (68.4)

Improvement, n (%)

3 (5.3) *

6 (11.7)

5 (8.8)

Overall progress of lung disease**

0.010

0.019

Improvement, n (%)

3 (5.3) *

7 (13.7)

6 (10.5)

Stabilization, n (%)

28 (48.4) *

34 (66.6)

35 (61.4)

Worsening, n (%)

26 (45.6) *

10 (17.5)

13 (22.8)

Death, n (%)

-

-

3 (5.3)

Inflammatory activity

DAS28, mean (SD)

3.8 (1.5)

3.0 (1.6)

3.1 (1.2)

0.019

0.024

C-reactive protein (mg/dl), median (IQR)

7.0 (5.0-17.7)

3.6 (2.4-10.1)

3.3 (2.6-7.9)

0.041

0.018

ESR (mm/h), median (IQR)

33.0 (15.0-47.0)

19.0 (6.0-32.0)

20.0 (8.0-29.0)

0.045

0.046

HAQ, median (IQR)

1.0 (0.2-1.8)

1.0 (0.3-1.8)

1.1 (0.6-1.9)

0.629

0.424

Corticosteroids, n (%)

39 (68.4)

30 (58.8)

28 (48.4)

0.360

0.040

Abbreviations: RA: rheumatoid arthritis; ILD: interstitial lung disease; DMARD: disease-modifying antirheumatic drug; SD: standard deviation; IQR: interquartile range; FVC: forced vital capacity; FEV1: forced expiratory volume in the first second; DLCO: diffusing capacity of the lung for carbon monoxide; UIP: usual interstitial pneumonia; NSIP: nonspecific interstitial pneumonia; HRCT: high-resolution computed tomography; DAS28: 28-joint Disease Activity Score; CRP: C-reactive protein; ESR: erythrocyte sedimentation rate; HAQ: Health Assessment Questionnaire; * Data from baseline vs date of diagnosis of ILD; ** Progression of lung disease overall taking into account HRCT and pulmonary function testing (FCV and DLCO). Pulmonary function tests considered progression (decrease in FVC >10% or in DLCO >15), stabilization (no progression or increase in FVC <10% or in DLCO <15%) and improvement (FVC ≥10% or DLCO ≥15%). HRCT considered progression (extension ≥ 20%), stabilization (extension < 20%) and improvement (decrease in extension).

.

-Pag 3; line 113-129: Working definitions and variables: “The main variable was the effectiveness of abatacept according to the outcome of ILD at the end of follow-up (FV) with respect to the following: (1) improvement (i.e., improvement in forced vital capacity [FVC] ≥10% or in the diffusing capacity of the lung for carbon monoxide [DLCO] ≥15% and no radiological progression); (2) non progression (stabilization or improvement in FVC <10% or in DLCO <15% and no radiological progression); (3) progression (worsening of FVC >10% or DLCO >15% or radiological progression); and (4) death.

Similarly, as secondary variables, we evaluated the change in FVC and DLCO, and radiologic progression at 12 months of treatment with abatacept (V12) and at the end of follow-up (FV). The lung function was considered progression (decrease in forced vital capacity [FVC] >10% or in the diffusing capacity of the lung for carbon monoxide [DLCO] >15), stabilization (no progression or increase in FVC <10% or in DLCO <15%) and improvement (FVC ≥10% or DLCO ≥15%) (31, 32). Radiologic evaluation was considered progression (≥20% increase in the presence and extension of ground-glass opacities, reticulation, honeycombing, diminished attenuation, centrilobular nodules, other nodules, emphysema, and consolidation), stabilization (extension <20%) and improvement (decrease in extension) (33-35).”

  1. ESR is definitely not a specific inflammatory markers and DAS28-CRP is favored in most studies. Given also the positive effect you show in the CRP change after abatacept initiation, I would consider using DAS28-CRP in the whole study. Still many patients 

Reply: There is debate about whether these two indices have any equivalence, and several studies have demonstrated that DAS28-CRP underestimates disease activity and overestimates response to treatment (Clin Exp Rheumatol 2008; 26, 769–75. / Ann Rheum Dis 2015; 74, 1132–7. / Ann Rheum Dis 2007; 66, 407–9 . / Joint Bone Spine 79, 149–55). In addition, a limitation in the CRP for the calculation of DAS28 is the use of laboratory kits that do not detect values below a certain threshold value of normality that varies from one laboratory to another instead of ultrasensitive CRP.

We have used DAS28-ESR because it is the measure that we defined to evaluate the inflammatory activity in the database in all the centers included when the study began. We think this is a good validated measure to assess inflammatory activity in RA patients. As the reviewer says, CRP and ESR data are also available independently. Although the evaluation of the number of painful and swollen joints as well as the analog scale were evaluated for the collection of the DAS28-ESR on the dates described in the protocol. However, we will take your comment into account for future studies to be able to include both measures, DAS28-ESR and DAS28-CRP.

Table 2. Progress of symptoms and lung disease at the end of follow-up in 57 patients with RA-ILD treated with abatacept.

Variable

Baseline

12 months

End of follow-up

p-value

Baseline vs 12 months

p-value Baseline vs End of follow-up

Follow-up in months, median (RIC)

-

12.6 (11.9-12.8)

27.3 (12.2-42.8)

-

-

Lung function

Oxygen saturation, mean (SD)

95.9 (2.4)

95.3 (3.6)

95.1 (3.9)

0.340

0.171

Pulmonary function tests

FVC predicted (%), mean (SD)

76.5 (18.2)

76.3 (16.1)

75.7 (14.4)

0.990

0.883

FVC <80%, n (%)

24 (42.1)

22 (43.1)

27 (47.3)

0.690

0.202

FVC ≥80%, n (%)

33 (57.9)

29 (56.8)

30 (52.6)

0.690

0.202

FEV1 predicted (%), mean (SD)

85.9 (10.4)

85.1 (19.1)

84.7 (13.9)

0.560

0.364

DLCO-SB predicted (%), mean (SD)

62.5 (16.5)

60.6 (16.0)

59.7 (15.7)

0.340

0.133

Clinical course

Progression, n (%)

25 (43.8) *

10 (19.6)

13 (22.8)

0.014

0.036

Stabilization, n (%)

29 (50.8) *

34 (66.6)

38 (66.7)

Improvement, n (%)

3 (5.3) *

7 (13.7)

6 (10.5)

HRCT

Radiological type

0.540

0.297

UIP, n (%)

34 (59.6)

31 (60.7)

36 (63.2)

NSIP, n (%)

18 (31.6)

16 (31.3)

16 (28.1)

Fibrotic NSIP, n (%)

5 (8.8)

4 (7.8)

5 (8.8)

Clinical course

0.019

0.028

Progression, n (%)

20 (35.0) *

10 (19.6)

13 (22.8)

Stabilization, n (%)

34 (59.6) *

35 (68.6)

39 (68.4)

Improvement, n (%)

3 (5.3) *

6 (11.7)

5 (8.8)

Overall progress of lung disease**

0.010

0.019

Improvement, n (%)

3 (5.3) *

7 (13.7)

6 (10.5)

Stabilization, n (%)

28 (48.4) *

34 (66.6)

35 (61.4)

Worsening, n (%)

26 (45.6) *

10 (17.5)

13 (22.8)

Death, n (%)

-

-

3 (5.3)

Inflammatory activity

DAS28, mean (SD)

3.8 (1.5)

3.0 (1.6)

3.1 (1.2)

0.019

0.024

C-reactive protein (mg/dl), median (IQR)

7.0 (5.0-17.7)

3.6 (2.4-10.1)

3.3 (2.6-7.9)

0.041

0.018

ESR (mm/h), median (IQR)

33.0 (15.0-47.0)

19.0 (6.0-32.0)

20.0 (8.0-29.0)

0.045

0.046

HAQ, median (IQR)

1.0 (0.2-1.8)

1.0 (0.3-1.8)

1.1 (0.6-1.9)

0.629

0.424

Corticosteroids, n (%)

39 (68.4)

30 (58.8)

28 (48.4)

0.360

0.040

-Pag 6; line 221-224: “Compared with initiation of abatacept, joint inflammation improved significantly at the end of follow-up in terms of DAS28-ESR (mean [DS], 3.1 [1.2] vs 3.8 [1.5] mg/l; p=0.024), CRP (median [IQR], 3.3 [2.6-7.9] vs 7.0 [5.0-17.7] mg/l; p=0.018), and ESR (median [IQR], 20.0 [8.0-29.0] vs 33.0 [15.0-47.0] mg/l; p=0.046).”

  1. selection for multivariate Cox using univariate has left corticosteroids dosage out of the model, although it should have entered.

Reply: According to the reviewer, we have included in the multivariate analysis the variables that were significant in the univariate analysis and those with clinical interest. The independent variables included in the equation were: sex, age, FVC, DAS28-ESR, DLCO-SB, methotrexate in combination, corticosteroids. However, in the final multivariate model, corticosteroids were not a significant variable. The sample size may not be sufficient to show significance in multivariate analysis. We have added this in the discussion.

-Pag 11; line 309-315: Table 4. Multivariate analysis of factors associated with progression of lung disease and mortality in patients with RA-ILD at initiation of treatment with abatacept. Cox regression model (adjusted for time receiving abatacept).

Variable

Univariate HR (95%CI)

Multivariate HR (IC 95%)

p-value

Age, years

1.000 (0.974-1.028)

Male sex

2.975 (0.618-7.109)

Current or previous smoking history

1.292 (0.386-4.321)

ACPA (>20 U/ml)

0.417 (0.096-1.809)

UIP radiological pattern

4.480 (0.896-12.390)

DAS28-ESR

2.001 (1.194-3.354)

2.520 (1.038-3.120)

0.041

FVC predicted (%)

0.879 (0.819-0.944)

0.826 (0.704-0.969)

0.019

DLCO-SB predicted (%)

0.889 (0.835-0.946)

0.837 (0.723-0.969)

0.018

Combination with methotrexate

0.267 (0.066-0.996)

Corticosteroids (grams)

1.272 (1.047-1.545)

Abbreviations. RA; rheumatoid arthritis; ILD: interstitial lung disease; ACPA: anticitrullinated peptide antibody; UIP: usual interstitial pneumonia; DAS28-ESR: 28-joint Disease Activity Score with erythrocyte sedimentation rate; FVC: forced vital capacity; DLCO-SB: diffusing capacity of the lung for carbon monoxide; Independent variables included in the equation at V0: sex, age, FVC, DAS28-ESR, DLCO-SB, methotrexate in combination, corticosteroids.

Nagelkerke R2: 0.47.

-Pag 12; line 363-366: “In relation to treatment, although corticosteroids were associated with progression of lung disease in the univariate analysis, it was finally not associated in the multivariate analysis. The sample size may not have been sufficient to show this association.”

To conclude, some minor comments:

  1. abstract reports enrollment between 2015-2020, the text says multiple times 2015-2021, which would not be sufficient to allow the minimum 12 months evaluation.

Reply: Between March 2015 and September 2021, a total of 59 patients with RA-ILD in prospective follow-up initiated abatacept, although 57 patients with a follow-up of ≥6 months were eventually included (Figure 1). The median (IQR) time with abatacept was 27.3 (12.2-42.8) months. There were only 6 patients with > 6 months of follow-up and less than 12 months. These 6 patients were excluded from the analysis at 12 months. We appreciate the reviewer's correction of the abstract. The authors apologize for this mistake in the abstract. We have modified the abstract and have clarified Patient flowchart in the manuscript.

-Pag 1; line 28-30: abstract: “Methods: We performed a prospective observational multicenter study of a cohort of patients with RA-ILD treated with abatacept between 2015 and 2021.”

-Pag 4; line 173-175: “Between March 2015 and September 2021, a total of 59 patients with RA-ILD in prospective follow-up initiated abatacept, although 57 patients with a follow-up of ≥6 months were eventually included (Figure 1). The median (IQR) time with abatacept was 27.3 (12.2-42.8) months.”

Pag. 13; line 390-393: “In our study there were 6 patients with more 6 months of follow-up but less than 12 months. These 6 patients were excluded from the analysis at 12 months. However, our main objective has been the pulmonary evolution at the end of the follow-up, and we also have most of the patients with data at 12 months to study their evolution.”

  1. reference 5 is not a systematic screening paper.

Reply: We appreciate the reviewer's comment and have corrected the bibliography.

-Pag 2, line 52-54: While around 20%-30% of patients develop clinically significant rheumatoid arthritis–associated interstitial lung disease (RA-ILD), systematic screening has shown that up to 35%-50% of patients with established RA develop the disease (5).

-Reference: 5.  Bilgici A, Ulusoy H, Kuru O, Celenk C, Unsal M, Danaci M. Pulmonary involvement in rheumatoid arthritis. Rheumatol Int. 2005;25(6):429-35.

  1. signals for stabilization of ILD has been seen previously also in other condition worth being mentioned (10.1016/j.semarthrit.2019.12.004)

Reply: Following the reviewer's comment, we have added this suggestion.

-Pag.11; line 322-323: “Stabilization of ILD with abatacept has been seen previously also in other condition such as in systemic sclerosis (44).”

-Reference 44: Castellví I, Elhai M, Bruni C, Airò P, Jordan S, Beretta L, et al. Safety and effectiveness of abatacept in systemic sclerosis: The EUSTAR experience. Semin Arthritis Rheum. 2020;50(6):1489-93.

  1. 80% abnormal cut-off for FVC and DLco requires reference

Reply: According to the reviewer we have added the reference 80% abnormal cut-off for FVC and DLco.

-Pag 3; line 135-138: “PFT included full spirometry, where the results were expressed as percent predicted adjusted for age, sex, and height. Abnormal FVC was defined as <80% predicted. DLCO was evaluated using the single-breath method (DLCO-SB), with a value of <80% considered abnormal (37).”

-Reference: 37. Lee YS, Kim HC, Lee BY, Lee CK, Kim MY, Jang SJ, et al. The Value of Biomarkers as Predictors of Outcome in Patients with Rheumatoid Arthritis-Associated Usual Interstitial Pneumonia. Sarcoidosis Vasc Diffuse Lung Dis. 2016;33(3):216-23.

  1. Figure 1: I assumed follow up time would have been minimum 12 month. How many other patients excluded for the other exclusion criteria?

Reply: We determined at least 6 months as inclusion criteria. Only 2 patients were excluded because they had been in treatment for less than 6 months. No more patients were excluded. There were only 6 patients with > 6 months of follow-up and less than 12 months. These 6 patients were excluded from the analysis at 12 months. But most of the patients had evolution at 12 months. So we believe that this could be interesting to see the evolution of the majority of the sample.

-Pag 4; line 173-175: “Between March 2015 and September 2021, a total of 59 patients with RA-ILD in prospective follow-up initiated abatacept, although 57 patients with a follow-up of ≥6 months were eventually included (Figure 1). The median (IQR) time with abatacept was 27.3 (12.2-42.8) months.”

Pag. 13; line 390-393: “In our study there were 6 patients with more 6 months of follow-up but less than 12 months. These 6 patients were excluded from the analysis at 12 months. However, our main objective has been the pulmonary evolution at the end of the follow-up, and we also have most of the patients with data at 12 months to study their evolution.”

  1. Figure 3: follow-up time/months should present a data in all histograms, for the fourth bar.

Reply: Thanks for the comment. We have added follow-up time/months in the figure caption or the fourth bar. In this way the figure is not very charged.

Figure 3. Progress of pulmonary function tests. Figure 3 shows the data for the pulmonary function tests at diagnosis of ILD, at the baseline visit (V0), at 12 months (V12), and at the final visit (FV): A) forced vital capacity; B) forced expiratory volume in the first second, C) diffusing capacity of the lung for carbon monoxide. The p value for baseline (V0) is the result of comparing values between V0 and the date of diagnosis of ILD. The p value for 12 months (V12) and at the end of follow-up (FV) is the result of the comparison between V12 and FV and baseline (V0). The median (IQR) follow-up time was 27.3 (12.2-42.8) months.

Thank you in advance for your time and consideration.

Sincerely yours,

*Correspondence:

Corresponding Author: Natalia Mena Vazquez. Instituto de Investigación Biomédica de Málaga (IBIMA). UGC de Reumatología, Hospital Regional Universitario de Málaga, Málaga, Spain.CP:29010. T: 952290360, E: [email protected]

Reviewer 2 Report

The authors describe the effects of abatacept in 57 patients with RA-ILD over a median of 27 months and report in detail on the influence of this agent on the associated ILD using well-defined criteria. They compare their results to others in the literature. The study is well conducted but I have a few comments:

1 Within the definitions of improved vs stablised vs deteriorated, how did the authors classify those whose lung function was stable but HRCT deteriorated and vice versa? The description given does not clarify this.

2 Although'only'6 (10%) of patients were hospitalised with a serious adverse event, nearly half did develop infection over a relatively short period. Although this is possibly less than for other biologics, the authors do not comment on the potential influence of steroids in these patients. Were those hospitalised with infection on higher steroid doses?

3 Among those 46% of patients whose disease had previously progressed prior to abatacept, was the outcome worse than those whose disease had been previously stable?  Although this may be inferred from the relationships between worse baseline VC / TLco and progression during therapy, it would be helpful to see this specifically addressed

4 A minor point: I find the sentence on Page 6 from line 203-205 somewhat confusing. Can this be clarified please.

Author Response

(The authors gave the same response as above.)

Reviewer 3 Report

Conclusions clearly presented. The study can be surely used for further extended researches

Author Response

(The authors gave the same response as above.)

Reviewer 4 Report

The manuscript presents the results of a small cohort study of RA-ILD patients treated with Abatacept. The nature of the study presents with several limitations, which should be better highlighted and do now allow to draw real conclusions about a possible efficacy of abatacept.

In addition, I would consider multiple major methodological limitation, which must be addressed to improve the overall quality of the manuscript:

- FV being in 2021 would be the same timepoint for patients enrolled in 2020; I do not see the real benefit of this timepoint overall, as 25% of patients have a FU duration below 12.2 years.

- the comparison between ILD diagnosis and Baseline for all parameters is not really informative, given the very wide range of ILD duration. The comparison does not take into account the time between the two assessments and, therefore, may be strongly unbalanced by the patients with twice longer disease duration.

- the 12 months evaluation is the real timepoint to consider in the study; what about comparing the change in FVC-DLCO between v0-v1 versus the change in the same between 12 months before starting abatacept and v0? That would show signal for different effect of abatacept and allow some comparison versus non-abatacept treatment.

- the combined outcomes proposed are very weak. in particular, where were the cut-offs derived from a patients who se FVC il 8% higher has not improved? combining DLco decline and HRCT progression? How was ILD progression on HRCT determined? according to which visual scoring assessment? increase in 20% is absolute or relative? A 20% absolute increase in ILD would represent a severe progressor and I would not expect the study to be able to pick up many events; this also implies that a 19% increase in ILD extent is not a radiologic progression? Hard to believe. No data at all is presented regarding ILD extent anyway, both at baseline and FU --> I would suggest considering changes in FVC alone, with values somehow close to the MCID proposed for IPF (du Bois RM, Weycker D, Albera C, et al. Forced vital capacity in patients with idiopathic pulmonary fibrosis: test properties and minimal clinically important difference. Am J Respir Crit Care Med 2011; 184: 1382–1389) otherwise take it from another CTD-ILD (10.1164/rccm.201709-1845OC). Otherwise, use the Goh et al 2017 definition of progression combining FVC and DLCO changes. 

- table 2 should present also all data at 12months, as it is one of the study timepoints. The difference between "clinical course" and "overall clinical course" is very unclear, as not specified in the methods section. is the first one only based on PFT?

- ESR is definitely not a specific inflammatory markers and DAS28-CRP is favored in most studies. Given also the positive effect you show in the CRP change after abatacept initiation, I would consider using DAS28-CRP in the whole study. Still many patients 

- selection for multivariate Cox using univariate has left corticosteroids dosage out of the model, although it should have entered.

To conclude, some minor comments:

- abstract reports enrollment between 2015-2020, the text says multiple times 2015-2021, which would not be sufficient to allow the minimum 12 months evaluation.

- reference 5 is not a systematic screening paper.

- signals for stabilization of ILD has been seen previously also in other condition worth being mentioned (10.1016/j.semarthrit.2019.12.004)

- 80% abnormal cut-off for FVC and DLco requires reference

- Figure 1 : I assumed follow up time would have been minimum 12 month. How many other patients excluded for the other exclusion criteria?

- Figure 3: follow-up time/months should present a data in all histograms, for the fourth bar.

Author Response

(The authors gave the same response as above.)

Round 2

Reviewer 4 Report

Thanks to the authors for addressing my comments.

Still details of how ILD on HRCT was quantified should be given: which method/score for determining %? Your references 34-35 refer to a visual and an a software method, respectively. What did you use?

Author Response

Comments for the reviewers

We would like to thank the editor for considering our work for publication in “Biomedicines” and the reviewers for their comments, which have helped to improve the quality of our manuscript.

Below, we provide a point-by-point reply to the comments.

Reviewer #4: 

Thanks to the authors for addressing my comments.

  1. Still details of how ILD on HRCT was quantified should be given: which method/score for determining %? Your references 34-35 refer to a visual and an a software method, respectively. What did you use?

Reply:  Yes, we have performed a visual method to evaluate the extension of ILD. The parenchymal pattern score was calculated based on the percentage ratio of the area of each parenchymal pattern to the total lung parenchyma. All HRCT scans were based on an axial slice measuring 1.5 or 2 mm in thickness taken at intervals of 1 cm along the thorax. Images were reconstructed using a high-spatial-frequency-algorithm, with 20 to 25 slices acquired per patient per minute. In order to homogenize the interpretation of findings, the radiological evaluation of the centers was centralized at HRUM and performed blind and independently by 2 experts in pulmonary radiology. In addition, the same 2 experts in pulmonary radiology that evaluated the patients during the prospective follow-up. This procedure was easily carried out because the majority of the centers share the same intranet. Only two centers are not part of this intranet, Vigo and Getafe, but their HRCT tests were sent at HRUM. We have specified this in the document.

Pag. 3; line 105-110: “All HRCT scans were based on an axial slice measuring 1.5 or 2 mm in thickness taken at intervals of 1 cm along the thorax. Images were reconstructed using a high-spatial-frequency-algorithm, with 20 to 25 slices acquired per patient per minute. we have performed a visual method to evaluate the extension of ILD and the parenchymal pattern score was calculated based on the percentage ratio of the area of each parenchymal pattern to the total lung parenchyma (31-33). In order to homogenize the interpretation of findings, the radiological evaluation was centralized at HRUM and performed blind and independently by 2 experts in pulmonary radiology.”

Pag. 3; line 130-134: “Radiologic evaluation was considered progression (≥20% increase in the presence and extension of ground-glass opacities, reticulation, honeycombing, diminished attenuation, centrilobular nodules, other nodules, emphysema, and consolidation), stabilization (extension <20%) and improvement (decrease in extension)(31-33, 36).

Reference:

  1. Ito Y, Arita M, Kumagai S, Takei R, Noyama M, Tokioka F, et al. Radiological fibrosis score is strongly associated with worse survival in rheumatoid arthritis-related interstitial lung disease. Mod Rheumatol. 2019;29(1):98-104.
  2. Kelly CA, Saravanan V, Nisar M, Arthanari S, Woodhead FA, Price-Forbes AN, et al. Rheumatoid arthritis-related interstitial lung disease: associations, prognostic factors and physiological and radiological characteristics--a large multicentre UK study. Rheumatology (Oxford). 2014;53(9):1676-82.
  3. Goh NS, Desai SR, Veeraraghavan S, Hansell DM, Copley SJ, Maher TM, et al. Interstitial lung disease in systemic sclerosis: a simple staging system. Am J Respir Crit Care Med. 2008;177(11):1248-54.

Thank you in advance for your time and consideration.

Sincerely yours,

*Correspondence:

Corresponding Author: Natalia Mena Vazquez. Instituto de Investigación Biomédica de Málaga (IBIMA). UGC de Reumatología, Hospital Regional Universitario de Málaga, Málaga, Spain.CP:29010. T: 952290360, E: [email protected]
